# Short-Term Effect of Air Pollution on Tuberculosis Based on Kriged Data: A Time-Series Analysis

**DOI:** 10.3390/ijerph17051522

**Published:** 2020-02-27

**Authors:** Shuqiong Huang, Hao Xiang, Wenwen Yang, Zhongmin Zhu, Liqiao Tian, Shiquan Deng, Tianhao Zhang, Yuanan Lu, Feifei Liu, Xiangyu Li, Suyang Liu

**Affiliations:** 1Hubei Provincial Center for Disease Control and Prevention, Wuhan 430079, China; hsq7513@163.com (S.H.); wenwenyanglinyi@163.com (W.Y.); 2Department of Global Health, School of Health Sciences, Wuhan University, 115 Donghu Road, Wuhan 430071, China; xianghao@whu.edu.cn (H.X.); 2018103050011@whu.edu.cn (F.L.); lxy329880@163.com (X.L.); 3Global Health Institute, Wuhan University, 115 Donghu Road, Wuhan 430071, China; 4College of Information Science and Engineering, Wuchang Shouyi University, Wuhan 430064, China; zhongmin.zhu@whu.edu.cn; 5State Key Laboratory of Information Engineering in Surveying, Mapping and Remote Sensing, Wuhan University, Wuhan 430079, China; tianliqiao@whu.edu.cn (L.T.); dyv587@foxmail.com (S.D.); tianhaozhang@whu.edu.cn (T.Z.); 6Environmental Health Laboratory, Department of Public Health Sciences, University of Hawaii at Manoa, 1960 East-West Road, Biomed Building, D105, Honolulu, HI 96822, USA; yuanan@hawaii.edu

**Keywords:** tuberculosis, infectious disease, air pollution, time-series, Poisson regression, kriging

## Abstract

Tuberculosis (TB) has a very high mortality rate worldwide. However, only a few studies have examined the associations between short-term exposure to air pollution and TB incidence. Our objectives were to estimate associations between short-term exposure to air pollutants and TB incidence in Wuhan city, China, during the 2015–2016 period. We applied a generalized additive model to access the short-term association of air pollution with TB. Daily exposure to each air pollutant in Wuhan was determined using ordinary kriging. The air pollutants included in the analysis were particulate matter (PM) with an aerodynamic diameter less than or equal to 2.5 micrometers (PM_2.5_), PM with an aerodynamic diameter less than or equal to 10 micrometers (PM_10_), sulfur dioxide (SO_2_), nitrogen dioxide (NO_2_), carbon monoxide (CO), and ground-level ozone (O_3_). Daily incident cases of TB were obtained from the Hubei Provincial Center for Disease Control and Prevention (Hubei CDC). Both single- and multiple-pollutant models were used to examine the associations between air pollution and TB. Seasonal variation was assessed by splitting the all-year data into warm (May–October) and cold (November–April) seasons. In the single-pollutant model, for a 10 μg/m^3^ increase in PM_2.5_, PM_10_, and O_3_ at lag 7, the associated TB risk increased by 17.03% (95% CI: 6.39, 28.74), 11.08% (95% CI: 6.39, 28.74), and 16.15% (95% CI: 1.88, 32.42), respectively. In the multi-pollutant model, the effect of PM_2.5_ on TB remained statistically significant, while the effects of other pollutants were attenuated. The seasonal analysis showed that there was not much difference regarding the impact of air pollution on TB between the warm season and the cold season. Our study reveals that the mechanism linking air pollution and TB is still complex. Further research is warranted to explore the interaction of air pollution and TB.

## 1. Introduction

Tuberculosis (TB) is one of the top ten deadly diseases in the world [1]. In 2016, there were about 10.4 million people diagnosed with TB worldwide, and 1.7 million people died from the disease [1]. Seven developing countries, China, India, Indonesia, Nigeria, the Philippines, Pakistan, and South Africa, account for 64% of the total cases [1]. As one of the countries with the highest number of TB cases, China has been working hard to lower the incidence of TB and has successfully reduced the TB-related mortality by 80% from 1990 to 2010 [2]. Despite this progress, China still has an estimated 1 million new cases each year [3] and faces serious drug-resistant TB epidemics [4]. China’s estimated TB incidence rate was 63/100,000 in 2017, ranking the 28th globally [3]. In Hubei Province where Wuhan is located, the reported incidence of TB in 2017 was 68.3/100,000, ranking ninth in the nation [5]. As the capital of Hubei Province and the largest city in central China, Wuhan also has a high incidence of TB. In 2017, 5952 new cases of tuberculosis were reported in Wuhan, with an estimated incidence of about 54.64/100,000 [5].

Previous studies have identified air pollution as one of the possible risk factors for TB. Most of these studies have focused on indoor exposure, which has been well summarized in several review articles [6,7,8,9]. These articles reveal that the use of biomass fuels (e.g., scrap lumber, crops, and manure) and fossil fuels (e.g., coal and diesel) are two environmental risk factors for TB infection. Tremblay (2007) studied the relationship between coal consumption and TB disease using historical statistics and proposed a hypothesis that the combustion of coal and possibly town gas aggravated the TB epidemics. Fullerton et al. (2008) claimed that the aggregated evidence from different studies across the world supports a causal relationship between biomass smoke exposure and the development of TB. Besides biomass and fossil fuels, tobacco smoking is also recognized as a non-communicable environmental risk factor for TB [10,11]. Compared with studies that have linked air pollution to TB in an indoor environment, fewer studies have focused on outdoor environments where the pollution level is relatively lower.

In epidemiologic studies that specifically investigated the associations between TB and ambient air pollution, the air pollutants associated with tuberculosis were not the same, and their effects varied across studies. Jassal et al. (2013) and You et al. (2016) found a positive association between exposure to ambient particulate matter (PM) with aerodynamic diameter of ≤ 2.5 μm (PM_2.5_) and the risk of TB. Zhu et al. (2018) observed a positive association of PM with an aerodynamic diameter of ≤ 10 μm (PM_10_) with the incidence of TB. Zhu et al. (2018) and Xu et al. (2019) found that gaseous pollutants sulfur dioxide (SO_2_) and nitrogen dioxide (NO_2_) were positively associated with the risk of TB, though the effects were modified by age and gender in the former study. In addition to those reported positive associations [12,13,14,15], other studies found no or weak associations [16,17], and one study even reported a protective effect of ambient SO_2_ on TB [18]. This inconsistency warrants further epidemiological studies on the topic. There are some possible mechanisms to explain the link between air pollution and the increased risk of TB. These mechanisms include (1) both gaseous pollutants and PM may trigger the disease by weakening the immune system (e.g., affecting T cells or impairing macrophage function); (2) oxidative stress and inflammatory reactions induced by air pollution may result in damage to the respiratory tract or lung epithelial cells; and (3) the direct transport of bacteria and those attached to particles infect the healthy population [19,20]. These mechanisms are discussed in more detail in the discussion section.

Researchers studying the health effects of ambient air pollution often rely on monitoring data that are routinely collected at ground environmental monitoring stations [21]. Typically, the collected data are averaged across the study area as a proxy measure to represent the exposure of the study population, resulting in measurement bias. Alternatively, spatial interpolation methods such as inverse distance weighting (IDW) and kriging are used to estimate ambient air pollution. Kriging can utilize the data collected at given positions (e.g., air monitoring stations) to predict the concentrations at unmeasured locations. In the past decades, kriging has been broadly used for mapping air pollution levels [22,23] and to estimate exposures in epidemiologic studies [24,25,26]. In the current study, we estimated air pollution levels using kriging and applied generalized additive models to assess associations between short-term exposure to ambient air pollutants and TB incidence in Wuhan city, China, during the 2015–2016 period.

## 2. Methods

### 2.1. Study Location

Wuhan, with a population of 10.6 million and an area of 8594 km^2^, is the largest city in central China and the capital city of Hubei Province. Wuhan has 13 administrative districts. As of 2014, Wuhan’s gross domestic product (GDP) per capita reached 15.6 thousand U.S. dollars, making it one of the fastest growing cities in China [27]. However, its urbanization and industrialization have exposed local residents to a high level of air pollution. For example, the annual average concentration of PM_2.5_ in Wuhan was 106.5 µg/m^3^ as of 2017, which was significantly higher than that of other cities with similar economic development levels [27]. Figure 1 shows the geographical location of Wuhan, provincially-designated air sampling stations, and population by administrative districts.

### 2.2. Air Pollution and Meteorological Data

The monitoring data used in the present study were collected by the Hubei Provincial Environmental Quality Supervision and Administration Bureau (http://www.hbemc.com.cn/). All air pollutants were measured based on a 24-h sampling schedule, except for ozone, which was collected using an 8-h schedule. In the current study, rather than calculating daily average concentrations for each air pollutant across multiple monitors as many studies usually do, we used kriging, a spatial interpolation method for exposure assessment, to estimate daily levels of ambient air pollutants including PM_2.5_, PM_10_, SO_2_, NO_2_, carbon monoxide (CO), and tropospheric ozone (O_3_) in Wuhan city. Although our interest was the Wuhan metropolitan area, we did include monitoring data from all sampling sites in Hubei Province to increase the statistical stability of our analysis. This led to a total of 51 monitoring sites included in the analysis, with 10 sites in Wuhan and an additional 41 sites across Hubei Province. We used ordinary kriging to determine the concentration of each air pollutant on each day for the Wuhan metropolitan area throughout the entire study period from 2015 to 2016. Specifically, kriging values generated from concentrations of air pollutants collected at the monitoring sites were first exported to a raster. The centroid of a given county or city was then used as the point data (represented by the geographical coordinates) to extract the interpolation results corresponding to the output raster. The analysis was repeated for each air pollutant on each day throughout the study period to obtain the complete exposure data set. It must be emphasized that kriging was used for predicting exposures, not TB incidence. The analysis was achieved using ArcMap 10.3 (ArcGIS. Version 10.3. Environmental Systems Research Institute, Redlands, CA, USA).

Daily meteorological data including temperature and relative humidity were collected at Wuhan Tianhe International Airport and obtained from https://rp5.ru/Weather_in_the_world.

### 2.3. TB Data

Daily TB incidence data were obtained from the Hubei Provincial Center for Disease Control and Prevention (Hubei CDC) for the 2015–2016 period. Hubei CDC collects information on a total of 39 infectious diseases that are legally regulated in China. Information on infectious diseases is collected at multiple levels, including health care institutions, individual practitioners, and rural doctors. Upon the collection of infectious disease information, designated epidemic reporters will diagnose infectious disease patients or suspected patients according to the latest diagnostic criteria for infectious diseases (Chinese Center for Disease Control and Prevention, 2008). In short, a chest X-ray is a common method for detecting tuberculosis, but the following tests are needed to confirm the diagnosis. First, check the patient’s sputum for *Mycobacterium tuberculosis* to confirm the diagnosis. Second, use a lung X-ray examination to diagnose the location, scope, and nature of the disease. Third, conduct a tuberculin test. A positive test indicates infection. County-level CDCs will review the information collected to ensure the quality of the data and then file it online in the Infectious Disease Reporting Information Management System. We extracted data from this online reporting system and used the International Classification of Disease, 10th revision (WHO, 2007), to identify TB cases (coded as A15.0–A15.3; A16.0–A16.2).

Ethics approval and consent to participate: The study was approved by the Ethics Committee of Wuhan University School of Medicine.

### 2.4. Statistical Analysis

We used ageneralized additive model (GAM) to relax the assumption of linearity between predictors and response variables. GAM is the most widely used method in air pollution epidemiology because it allows for nonparametric adjustment of nonlinear confounding effects of seasonality, trends, and weather variables. Variables such as temperature and humidity can be controlled in the model by specifying a spline, a function to smooth the parameters for predicting the optimizing results [28]. We used GAMs to examine the associations between daily incidence of TB and daily concentrations of air pollutants (PM_2.5_, PM_10_, SO_2_, NO_2_, CO, and O_3_) in Wuhan during the 2015–2016 period. We also examined the potential seasonal variation by stratifying the data set during warm (May–October) and cold (November–April) seasons. Typically, time-series data are “overly dispersed”, which means the variance in a data set exceeds the mean. To account for overdispersion, the daily TB counts were assigned by quasi-Poisson distribution, a common way to deal with over-dispersed data. We used natural cubic spline functions to adjust for the time-varying variables. Specifically, the daily average temperature and relative humidity were adjusted with 3 degrees of freedom (df). The long-term and seasonal effects were adjusted with 8 df per year of data. Also included in the model is day of week (DOW), which is a set of indicator variables shown on each day of the week. The selection of covariates and df were based on a previously published study [25]. To explore the lag structures between incident TB and each air pollutant, a single-day lag model was examined from lag 0 through lag 14. The “lag” is defined as the time between the reporting of a TB case and days of exposure prior to this reporting. To report the results, we calculated the relative risk (RR) of TB associated with a 10 μg/m^3^ increase in each pollutant concentration, with corresponding 95% confidence intervals (CIs).

To test the robustness of the results, we performed two sensitivity analyses. In the first sensitivity analysis, we used different df values for long-term effects (e.g., 4, 6, 10, and 12 df per year). In the second sensitivity analysis, we adopted different temperature and humidity values in our model (e.g., values at t-1 and the cumulative effect calculated by three-day moving averages) to test whether the results were different from those calculated using daily averages in the current model.

We used SAS (version 9.3; SAS Institute Inc., Cary, NC, USA) to combine TB data, meteorological data, and air pollution data, as well as generate descriptive statistics. Our time-series analysis used the R (version 3.0.2; http://R-project.org) statistical package “mgcv” to establish the model and calculate the results.

## 3. Results

Overall, there were 12,648 incident cases of TB included in our analysis during this two-year study. The cumulative incidence rate of TB was 1.3% in Wuhan during the 2015–2016 period. Table 1 shows the descriptive statistics for TB incident cases, air pollutants, and meteorological data. On average, there were 17 incident cases of TB per day. The average daily number of TB was the highest during spring season (March, April, and May), whereas the lowest was observed in winter months (December, January, and February). Air pollutant data were close to completeness, with only 9 days missing. Daily average concentrations of air pollutants were all higher during the cold season than in the warm season. The respective daily mean and standard deviation of PM_2.5_ concentration were 58.94 µg/m^3^ and 35.41 µg/m^3^, lower than those of PM_10_ (89.34 ± 42.16 µg/m^3^). The highest level of PM_2.5_ was observed in January (113.4 µg/m^3^) while the lowest was observed in July (29.7 µg/m^3^). For gaseous pollutants SO_2_, NO_2_, CO and O_3_, the daily mean concentrations were 15.32, 38.61, 1.03 and 119.23 µg/m^3^, respectively.

We fit a separate model to the relative risk (RR) of TB incident cases for each air pollutant at lag 0 through 14 and present the results in Figure 2. The single-day lag models show that the effect of air pollutants on TB was greatest at lag 7. Among them, PM_2.5_, PM_10_, and O_3_ were statistically significantly (*p* < 0.05) associated with TB. We therefore report our time-series analysis results based on exposure at lag 7 because of this lag structure. The effect of air pollutants on TB at lag 7 is presented in Table 2. For a 10 µg/m^3^ increase in PM_2.5_, PM_10_, and O_3_, the TB risk increased by 17.03% (95% CI: 6.39, 28.74), 11.08% (95% CI: 6.39, 28.74), and 16.15% (95% CI: 1.88, 32.42), respectively. We also fit a multi-pollutant model including all pollutants simultaneously to assess whether associations found in single-pollutant models were confounded by the presence of co-pollutants. After controlling for co-pollutants, the association of PM_2.5_ with TB was still robust and remained statistically significant. The effect estimates were slightly attenuated for all remaining pollutants in the multi-pollutant model.

The season-specific effects of each air pollutant on TB are presented in Table 2 and further illustrated in Figure 3. We did not observe an obvious trend in seasonality between the warm season (May–October) and the cold season (November–April). The associations between PM_2.5_ and TB were statistically significant (*p* < 0.05) for both warm and cold seasons. The effect of NO_2_ on TB during the cold season was also statistically significant (*p* < 0.05). It is noteworthy that dividing the data into warm and cold seasons reduced the sample size and led to an expanded confidence interval.

## 4. Discussion

In this study, we conducted a time-series analysis to estimate short-term effects of air pollution on TB incidence in Wuhan city during the 2015–2016 period. We found that the effect of air pollutants on TB was greatest at lag 7. PM_2.5_ and PM_10_, and O_3_ were statistically significantly associated with the increased risk of TB at the 7th day’s exposure. We fitted a multi-pollutant model to adjust for the possible confounding from the presence of co-pollutants. Our results showed that the effect of PM_2.5_ on TB remained robust after the adjustment while the effects for the rest of the pollutants were reduced or remained almost unchanged (O_3_). The seasonal analysis showed that the impact of air pollution on TB was not that different between the warm season and the cold season.

One noteworthy finding of this research is that the effect of air pollution on TB was greatest at lag 7. The surprising consistency observed in lag 7 may be due to the ubiquitous collinearity among the pollutants, that is, the effect of any pollutant on TB actually reflects the effect of another or a combination of several pollutants on TB. Our results are contrary to many previous studies in which the effect was larger in early lags (typically at lag 0–3). This is likely due to the incubation period of the TB bacterium *Mycobacterium tuberculosis*. Our finding agrees with a previously published article in which significant associations between respiratory infection and PM_10_/PM_2.5_ were observed at lag 9–10 days and 7–8 days, respectively [29]. Chen et al. (2017) also reported a lagged effect of PM_2.5_ on influenza and attributed this to the incubation of the influenza virus [30]. Considering that the incubation period of the TB bacterium varies from weeks to decades [18], it is possible that exposure to ambient air pollution shortened the incubation period and accelerated the progress of the disease. Many previous studies have provided a theoretical basis for this hypothesis. Animal studies and cellular-level studies suggest that biomass smoke impairs alveolar macrophage function [31,32,33]. PM_2.5_ contains heavy metals that may damage macrophages, resulting in weakened phagocytosis [33]. It has been suggested that air pollutants PM_2.5_, CO, NO_2_, and SO_2_ can modify the regulation of tumor necrosis factor-alpha (TNF-α) and interferon-gamma (IFN-*γ*) [9,33,34,35], which functions to induce macrophage activation [33,34]. Consequently, the damaged system may lead to an increased incidence of TB [6]. Additionally, studies have found that bacteria and viruses can attach to particles in the air and be inhaled into the respiratory system [36,37,38]. This makes the mechanism even more complex as one cannot judge if infection occurred before or during exposure to air pollution.

We observed a statistically significant association between PM_2.5_ and PM_10_ and TB in our analysis. We noted that another study in Wuhan also examined the relationship between short-term exposure to air pollution and the risk of TB [15]. However, our research differs from theirs in three ways. First, the two studies used different statistical models. We used GAM, and they used distributed lag non-linear models (DLNMs). Second, exposure to air pollution was measured differently. We used ordinary kriging to estimate the exposure, whereas they used data collected from air monitoring stations directly. Third, our study conducted seasonal analysis, and they did not. Possibly due to these differences, the results of the two studies are also different. They observed the effects of NO_2_, SO_2_, CO, and PM_2.5_ on the risk of TB, while we only observed the effects of PM on the risk of TB. In addition, their study had more years of data and was, therefore, easier to detect significant statistical associations, which could explain the differences in results between the two articles. Because no consensus has been reached on which combination of co-pollutants should be included in the multi-pollutant model [39], we included all air pollutants in the model. After the adjustment, the effect of PM_2.5_ on TB remained robust, while the effects of other pollutants were attenuated. However, the attenuated effects of other pollutants in the multi-contaminant model may be due to the collinearity between the pollutants, which does not necessarily mean that they have no effect on TB. This finding is consistent with previous epidemiologic studies on TB and ambient PM_2.5_ [12,13]. Nevertheless, the interpretation should be regarded with caution, as the presence of collinearity cannot be ignored to influence the association. Our results showed certain degrees of positive correlations between pairs of air pollutants (*r*: 0.59 to 0.80). Therefore, a single-pollutant model might be more reasonable for these types of data. We also observed positive but insignificant associations between gaseous air pollutants and TB. As mentioned earlier, CO, NO_2_, and SO_2_ all have roles in impairing the normal function of alveolar macrophages and result in an increased number of TB cases. Additionally, NO_2_ and SO_2_ can form ammonium sulfate and ammonium nitrate by naturally occurring chemical processes and can form part of PM and then further threaten human health [39].

Weather, especially temperature and humidity (or dew point temperature), are factors that can confound the relationship between health effects and air pollution at both the long- and short-term timescales. Temperature and humidity can be correlated with both air pollutants and health effects such as infectious diseases. We used GAM, a type of semiparametric model, to control for temperature and humidity. The semiparametric model combines the advantages of both parametric and nonparametric models, allowing for explicit parameter items that include exposures of interest and smooth nonparametric items such as temperature and humidity. The adjustment was achieved by using the smooth functions of temperature and humidity in the model. The smoothness of a model fit is controlled by the number of knots used, which was determined by the number of degrees of freedom assigned to each function. It has been reported that the potential confounding effect of temperature on short-term time scales may be less pronounced than long-term time scales, and the association between air pollution and health effects is relatively insensitive to the choice of the temperature model [40].

This article showed that TB incidence was highest in the spring. Despite that the underlying mechanism remains unknown, several factors may explain this phenomenon. First, people will reduce outdoor activities during the cold season. Consequently, the overcrowded, humid, and low airflow indoor environment makes TB easier to spread [41]. Meanwhile, more indoor activities reduce ultraviolet light exposure that can kill microorganisms [41]. Second, the decrease in human serum vitamin D in winter will increase the infection and re-activation of TB [42]. In addition, delays in seeking health care after the onset and diagnosis may also result in delayed reporting from the winter to the following spring [41]. This also led to misclassification of the actual exposure date. Our seasonal analysis suggested that the effects of air pollution on TB during the cold and warm seasons are not significantly different, which may mean that the increase in TB during spring is related to the indoor environment and is less associated with outdoor air pollution.

We examined the seasonal variation by splitting the all-year data into warm (May–October) and cold (November–April) seasons. There was no clear trend that the effect of air pollutants on TB was influenced by season. The effect estimates of SO_2_ and NO_2_ during the warm season were slightly higher than during the cold season. Certain factors may explain this phenomenon. First, during the warm season, people tend to go outdoors more often than during the cold season, so they are exposed to more ambient air pollutants as compared to the opposite. Second, more outdoor activities increase the possibility of TB infection, as elevated levels of airborne *Mycobacterium tuberculosis* may be present in the air. We also observed that the effect estimates of PM_2.5_, PM_10_, CO, and O_3_ during the warm season were slightly lower than during the cold season. This was consistent with findings of another study that the effect of ambient PM_2.5_ on TB was most significant during winter [13]. The study explained that staying indoors during high ambient air pollution episodes would lead to an increase in human contact and thus the increased risk of TB transmission. The same hypothesis was given elsewhere as well [43]. Therefore, the real mechanism is not yet clear and warrants more related research.

A feature of this study is the use of kriging to estimate the concentration of air pollution. Kriging has been broadly used to estimate ambient air pollution levels in epidemiological studies [22,24,25,44,45]. However, only a small part of the literature used kriging to study TB. Many such studies used kriging for the detection of TB clusters for animals and humans [46,47,48].

To the best of our knowledge, no studies have used the exposures estimated by kriging to conduct time-series research on TB. We attempted to use kriging to overcome a long-standing limitation existing in the epidemiology of air pollution, that is, to average the pollution level across the study area as the proxy for the population exposure level. By using kriging, we hope to reduce the misclassification raised by the simple averaging method. Nevertheless, it must be pointed out that kriging itself also has limitations. The accuracy of prediction using kriging largely depends on the number of monitoring stations. Due to a low ratio of monitoring stations to total land area, prediction error is quite possible. Furthermore, kriging also assumes isotropy, that is, it assumes uniformity in all directions. This obviously ignores the impact of the real environment (such as the built environment) on exposure [49]. A previous study compared the values estimated by kriging with the values estimated by the Assessment System for Population Exposure Nationwide (ASPEN) model and found that the former was generally smaller than the latter [21]. As estimates from the ASPEN model yielded a good agreement with monitoring data in multiple studies [21], it is considered more reliable than kriged values. In China, we do not have an air pollution prediction model similar to the ASPEN for comparison; thus, the accuracy of the prediction results is still uncertain.

### Limitations and Implications

Our study has several limitations. First, our ecological study design cannot capture exposure levels at an individual level, resulting in exposure misclassifications. Second, two years of data assume a small sample size and may reduce the statistical power for the detection of significance. Compared to studies with many years of data, this led to overly wide confidence intervals. Previous studies have reported that many air pollutants are statistically significantly associated with TB. However, we only found that there was a statistically significant association of PM_2.5_, PM_10_, and O_3_ with TB. Third, our O_3_ data were collected 8 h per day, while other pollutants were collected based on a 24-h sampling schedule. This will cause bias in our results. Fourth, due to the lack of data, some factors that can lead to confounding or effect modification, such as gender, age, socioeconomic status, and so on, have not been included in the analysis, which may result in bias in results. Fifth, because TB patients may not be able to seek health care immediately after they become ill, and it takes time from the diagnosis to the reporting of a case, the results of this article are affected by this unavoidable measurement error. Lastly, the existence of collinearity among air pollutants may create bias and affect the magnitude of the estimated associations. Considering the above limitations, caution is needed in interpreting the results of the article.

## 5. Conclusion

In summary, this study explored associations between short-term exposure to ambient air pollutants and TB incidence in Wuhan city, China, during the 2015–2016 period. We found that the impact of air pollutants on tuberculosis was greatest at the 7th day’s lag, and PM_2.5_, PM_10_, and O_3_ were significantly associated with an increased risk of TB. Considering that the health impacts of air pollution and infectious disease are typically studied separately, the findings of this study not only add to the literature regarding the short-term effect of air pollution on TB but also help us to understand the mechanism of the interaction of air pollution and infectious disease. Future preventive efforts against TB should consider the reduction of exposure to air pollution at the community or personal level.

## Figures and Tables

**Figure 1 ijerph-17-01522-f001:**
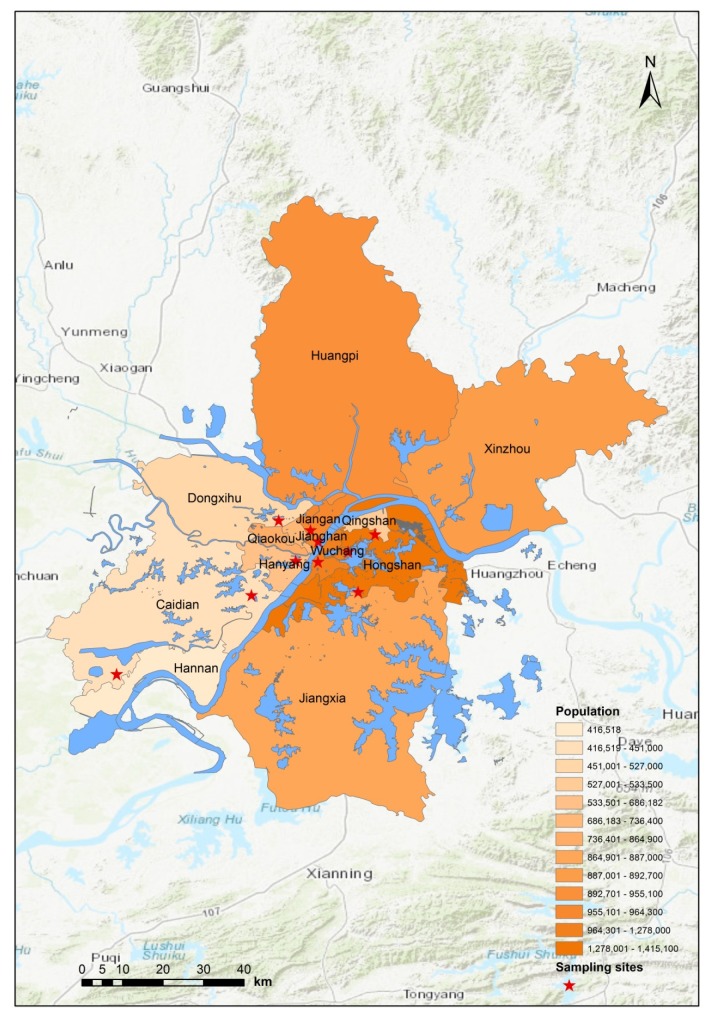
Map of Wuhan. The red pentagram represents monitoring stations. Population density by district is represented by gradient color.

**Figure 2 ijerph-17-01522-f002:**
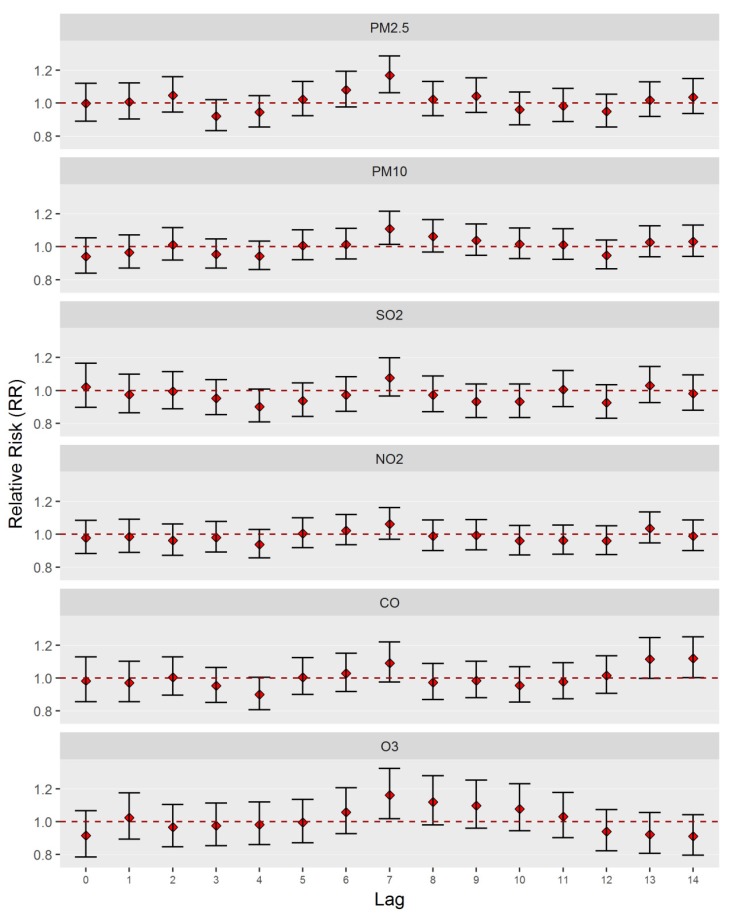
Relative risks (and 95% CIs) of TB incidence associated with an increase of 10 μg/m^3^ in air pollutant concentrations at lag 0–14 days in Wuhan, China, during the 2015–2016 period.

**Figure 3 ijerph-17-01522-f003:**
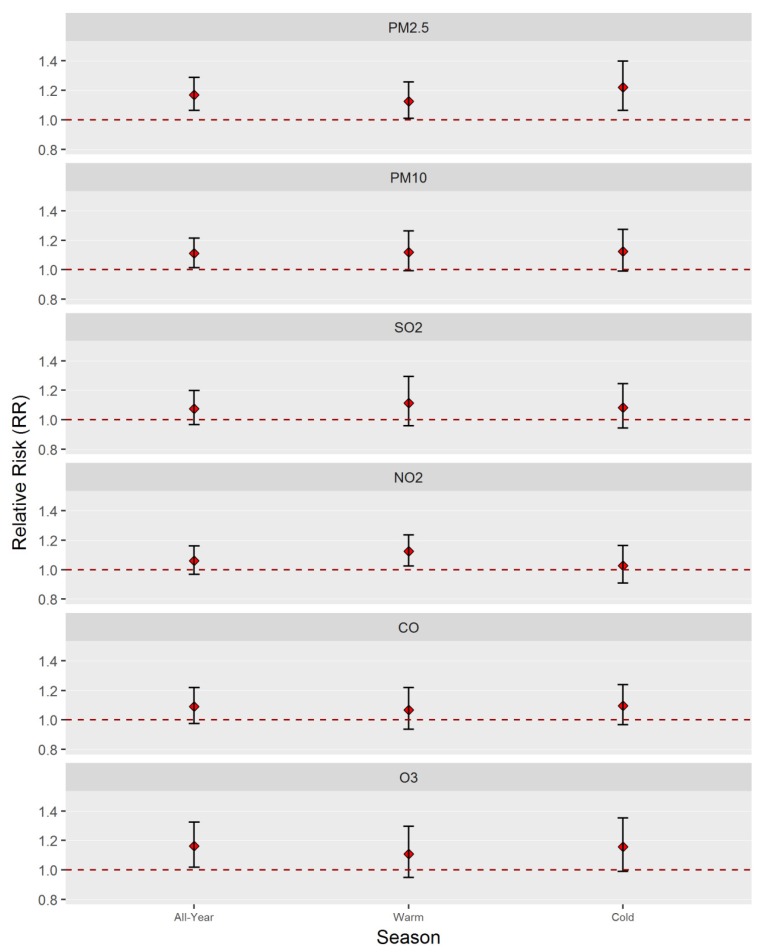
Relative risks (and 95% CIs) of TB incidence associated with an increase of 10 μg/m^3^ in air pollutant concentrations by season (all-year, warm season [May–October] and cold season [May–October]) at lag 7 in Wuhan, China, during the 2015–2016 period.

**Table 1 ijerph-17-01522-t001:** Descriptive statistics of TB incidence counts, air pollutants, and meteorological variables in Wuhan, China, during the 2015–2016 period.

Variable	Days	Mean ± SD			Percentiles ^a^				
		All-year	Warm ^b^	Cold	Min	25%	50%	75%	Max
Daily TB ^c^ cases	732	17.28 ± 13.49	17.67 ± 12.87	16.89 ± 14.10	2	11	14	19	134
Pollutant (µg/m^3^)									
PM_2.5_ ^d^	723	58.94 ± 35.41	40.93 ± 20.21	77.71 ± 38.04	8.31	33.60	49.03	76.84	227.20
PM_10_	723	89.34 ± 42.16	74.86 ± 33.47	104.42 ± 44.95	11.36	57.30	83.71	115.23	282.05
SO_2_	723	15.32 ± 7.80	11.66 ± 4.81	19.13 ± 8.47	3.28	9.74	13.50	18.97	53.53
NO_2_	723	38.61 ± 15.01	32.43 ± 11.65	45.06 ± 15.42	11.53	27.20	35.78	46.89	94.72
CO	723	1.03 ± 0.30	0.88 ± 0.20	1.18 ± 0.30	0.42	0.80	0.99	1.22	2.11
O_3_	731	119.23 ± 53.98	147.95 ± 51.49	89.96 ± 38.56	9.30	74.40	114.29	159.04	272.01
Weather variables									
Temperature (°F)	732	63.02 ± 16.02	75.92 ± 8.31	49.92 ± 10.29	26.08	48.29	65.56	76.79	91.36
Relative Humidity (%)	732	79.23 ± 10.80	79.55 ± 9.79	78.9 ± 11.74	41	72.25	80	87	100

^a^ Min = minimum, Max = Maximum; ^b^ Warm = May–October; Cold = November–April; ^c^ TB = Tuberculosis; ^d^ PM_2.5_ = particulate matter (PM) with an aerodynamic diameter less than or equal to 2.5 μm, PM_10_ = PM with an aerodynamic diameter less than or equal to 10 μm, SO_2_ = sulfur dioxide, NO_2_ = nitrogen dioxide, CO = carbon monoxide, O_3_ = ground-level ozone.

**Table 2 ijerph-17-01522-t002:** Increased risk of TB and corresponding 95% CIs associated with 10 µg/m^3^ increase in air pollutants ^a^ on the 7th lag day for single- and multi-pollutant models.

Pollutant	Single Pollutant Model			Multi-Pollutant Model ^b^
	All-Year	Warm Season ^c^	Cold Season	All-Year
PM_2.5_	**1.17 (1.06, 1.28) ^d^**	**1.13 (1.01, 1.26)**	**1.22 (1.06, 1.40)**	**1.20 (1.02, 1.41)**
PM_10_	**1.11 (1.01, 1.22)**	1.12 (0.99, 1.26)	1.12 (0.99, 1.27)	0.98 (0.84, 1.15)
SO_2_	1.08 (0.97, 1.20)	1.11 (0.96, 1.29)	1.08 (0.94, 1.24)	1.00 (0.86, 1.16)
NO_2_	1.06 (0.97, 1.16)	**1.13 (1.02, 1.24)**	1.03 (0.91, 1.16)	0.97 (0.84, 1.12)
CO	1.09 (0.97, 1.21)	1.07 (0.94, 1.22)	1.10 (0.97, 1.24)	0.96 (0.82, 1.12)
O_3_	**1.16 (1.02, 1.32)**	1.11 (0.95, 1.30)	1.16 (0.99, 1.35)	1.11 (0.95, 1.30)

^a^ PM_2.5_ = particulate matter (PM) with an aerodynamic diameter less than or equal to 2.5 μm, PM_10_ = PM with an aerodynamic diameter less than or equal to 10 μm, SO_2_ = sulfur dioxide, NO_2_ = nitrogen dioxide, CO = carbon monoxide, O_3_ = ground-level ozone; ^b^ The effect of each pollutant was adjusted for all the remaining pollutants; ^c^ Warm season = May–October; Cold season = November–April; ^d^ Statistically significant associations are shown in bold.

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
