# Peer review of "Short-Term Effect of Air Pollution on Tuberculosis Based on Kriged Data: A Time-Series Analysis"

_ijerph, 2020, doi:10.3390/ijerph17051522_

Round 1

Reviewer 1 Report

General comments:

It is very interesting and important to evaluate the relationship between air pollution and TB. The authors used an original statistical model (GAM) to illustrate the association, which resolve the nonlinear issues between pollutants and TB prevalence. Overall, the pathway to achieving the purpose and the analysis conclusion of this paper is clear and reasonable. It could be accepted after some minor revisions.

Specific comments:

1, Line 46: Valid data should be listed to point out that Wuhan is an area with high risks of TB.

2, Line 48: Air pollution has been identified as one of the possible risk factors for TB, what is the difference between the effects of different air pollutants for TB, compared with other studies?

3, In the background of the article, the author should point out the current status of air pollution in Wuhan. Furthermore, how about the incidence of tuberculosis in Wuhan compared with other cities in China or even in other countries?

4, This study didn’t consider the interference of covariates on the incidence of TB such as age, gender, etc.

6, Why did the authors divide the 12 months of the year into the warm and cold seasons instead of the spring, summer, autumn, and winter?

7, The reasons for the highest average daily morbidity of TB during spring season should be further discussed.

8, I think the authors should add a section – “Limitations and implications” at the end of the article.

Author Response

It is very interesting and important to evaluate the relationship between air pollution and TB. The authors used an original statistical model (GAM) to illustrate the association, which resolve the nonlinear issues between pollutants and TB prevalence. Overall, the pathway to achieving the purpose and the analysis conclusion of this paper is clear and reasonable. It could be accepted after some minor revisions.

Response: thank you for your support and encouragement for this article. Below are our responses to your comments.

Specific comments:

1, Line 46: Valid data should be listed to point out that Wuhan is an area with high risks of TB.

Response: We have added the relevant data in the manuscript and quoted below:

"China's estimated TB incidence rate was 63 / 100,000 in 2017, ranking the 28th globally [3]. In Hubei Province where Wuhan is located, the reported incidence of TB in 2017 was 68.3 / 100,000, ranking the ninth in the nation [5]. As the capital of Hubei Province and the largest city in central China, Wuhan also has a high incidence of TB. In 2017, 5,952 new cases of tuberculosis were reported in Wuhan, with an estimated incidence of about 54.64 / 100,000 [5]."

References:

Tuberculosis in China. Available online: http://wwwwprowhoint/china/mediacentre/factsheets/tuberculosis/en/ (Accessed on January 03, 2019). Reported incidence of tuberculosis in Hubei. Available online: http://www.hbcdc.cn/ (Accessed on January 03, 2019).

2, Line 48: Air pollution has been identified as one of the possible risk factors for TB, what is the difference between the effects of different air pollutants for TB, compared with other studies?

Responses: We have added a paragraph after the second paragraph of the introduction to explain the differences between the effects of air pollutants on tuberculosis in different studies and quoted below:

"In epidemiologic studies that specifically investigated the associations between TB and ambient air pollution, the air pollutants associated with tuberculosis were not the same, and their effects were varied across studies. Jassal et al. (2013) and You et al. (2016) found a positive association between exposure to ambient particulate matter (PM) with aerodynamic diameter of ≤ 2.5 μm (PM2.5) and risk of TB. Zhu et al. (2018) observed a positive association of PM with an aerodynamic diameter of ≤ 10 μm (PM10) with the incidence of TB. Zhu et al. (2018) and Xu et al. (2019) found that gaseous pollutants sulfur dioxide (SO2) and nitrogen dioxide (NO2) were positively associated with the risk of TB though the effects were modified by age and gender in the former study. In addition to those reported positive associations [13-16], other studies found no or weak associations [17-18], and one study even reported a protective effect of ambient SO2 on TB [19]. This inconsistency warrants further epidemiological studies on the topic."

References:

Jassal, M.S., I. Bakman, and B. Jones, Correlation of ambient pollution levels and heavily-trafficked roadway proximity on the prevalence of smear-positive tuberculosis. Public Health, 2013. 127(3): p. 268-274. You, S., et al., On the association between outdoor PM2. 5 concentration and the seasonality of tuberculosis for Beijing and Hong Kong. Environmental Pollution, 2016. 218: p. 1170-1179. Zhu, S., et al., Ambient air pollutants are associated with newly diagnosed tuberculosis: A time-series study in Chengdu, China. Science of the total environment, 2018. 631: p. 47-55. Xu, M., et al., Association of air pollution with the risk of initial outpatient visits for tuberculosis in Wuhan, China. Occupational and environmental medicine, 2019. 76(8): p. 560-566. Lai, T.-C., et al., Ambient air pollution and risk of tuberculosis: a cohort study. Occup Environ Med, 2016. 73(1): p. 56-61. Smith, G.S., et al., Particulate air pollution and susceptibility to the development of pulmonary tuberculosis disease in North Carolina: an ecological study. International journal of environmental health research, 2014. 24(2): p. 103-112. Ge, E., et al., Ambient sulfur dioxide levels associated with reduced risk of initial outpatient visits for tuberculosis: a population based time series analysis. Environmental Pollution, 2017. 228: p. 408-415.

3, In the background of the article, the author should point out the current status of air pollution in Wuhan. Furthermore, how about the incidence of tuberculosis in Wuhan compared with other cities in China or even in other countries?

Responses: Thank you, we have replied about the status of tuberculosis in Wuhan in question 1. Regarding the current status of air pollution, we have the following description.

"However, its urbanization and industrialization have exposed local residents to a great level of air pollution. For example, the annual average concentration of PM2.5 in Wuhan was 106.5 µg/m3 as of 2017, which was significantly higher than that of other cities with similar economic development levels [25]."

Reference:

Chu, Y., et al., Propensity to Migrate and Willingness to Pay Related to Air Pollution among Different Populations in Wuhan, China. Aerosol and Air Quality Research, 2017. 17(3): p. 752-760.

4, This study didn’t consider the interference of covariates on the incidence of TB such as age, gender, etc.

Responses: We are also aware that this is a limitation. However, we do not have such data and therefore cannot include them in our analysis. We listed it as a limitation in the discussion section, and quoted below.

"Fourth, due to the lack of data, some factors that can lead to confounding or effect modification, such as gender, age, socioeconomic status, and so on, have not been included in the analysis, which may result in bias in results."

6, Why did the authors divide the 12 months of the year into the warm and cold seasons instead of the spring, summer, autumn, and winter?

Responses: This is a very good question. We know that dividing the year-round data into four seasons will improve the accuracy of stratification. However, since we only have two years of data, doing so will cause us to lose a lot of statistical power and thus a wide confidence interval. So we decided to divide it into cold and warm seasons instead of four seasons.

7, The reasons for the highest average daily morbidity of TB during spring season should be further discussed.

Responses: We have added a paragraph to discuss this, quoted below.

"This article observed that TB incidence was highest in the spring. Despite the underlying mechanism is unknown, several factors may explain this phenomenon. First, people will reduce outdoor activities during the cold season. Consequently, the overcrowded, humid, and low airflow indoor environment makes TB easier to spread [26]. Meanwhile, more indoor activities reduce ultraviolet light exposure that can kill microorganisms [26]. Second, the decrease in human serum vitamin D in winter will increase the infection and re-activation of TB [27]. In addition, delays in seeking health care after the onset and diagnosis may also result in delayed reporting from the winter to the following spring [26]. This also led to misclassification of the actual exposure date. Our seasonal analysis suggested that the effects of air pollution on TB during the cold and warm seasons are not significantly different, which may mean that the increase in TB during spring is related to the indoor environment and less associated with outdoor air pollution."

References:

Fares, A., Seasonality of tuberculosis. Journal of global infectious diseases, 2011. 3(1): p. 46. Nnoaham, K.E. and A. Clarke, Low serum vitamin D levels and tuberculosis: a systematic review and meta-analysis. International journal of epidemiology, 2008. 37(1): p. 113-119.

8, I think the authors should add a section – “Limitations and implications” at the end of the article.

Responses: We added this section at the end of the discussion section, quoted below.

"4.1 Limitations and implications. Our study has several limitations. First, our ecological study design cannot capture exposure levels at individual level, resulting in exposure misclassifications. Second, two years of data assumes a small sample size and may reduce the statistical power for the detection of significance. Compared to studies with many years of data, this led to overly wide confidence intervals. Previous studies have reported that many air pollutants are statistically significantly associated with TB. However, we only found that there is a statistically significant association of PM2.5, PM10, and O3 with TB. Third, our O3 data were collected 8-hour per day, while other pollutants were collected based on 24-hour sampling schedule. This will cause bias in our results. Fourth, due to the lack of data, some factors that can lead to confounding or effect modification, such as gender, age, socioeconomic status, and so on, have not been included in the analysis, which may result in bias in results. Fifth, because TB patients may not be able to seek health care immediately after they become ill, and it takes time from the diagnosis to the reporting of a case, the results of this article are affected by this unavoidable measurement error. Lastly, the existence of collinearity among air pollutants may create bias and affect the magnitude of the estimated associations. Considering the above limitations, caution is needed in interpreting the results of the article."

Reviewer 2 Report

This is a well-described study. However, it lacks original/raw data to support its conclusion. Also, similar study has been done for 2014-2017 time period in Wuhan, China.

Author Response

This is a well-described study. However, it lacks original/raw data to support its conclusion. Also, similar study has been done for 2014-2017 time period in Wuhan, China.

Response: Thank you for your comment. This article uses a different method to assess the exposure to air pollution, which is called ordinary kriging. This is an innovation of this article. In the epidemiological investigation, the classic GAM is used in this article. Although this method is more commonly used for chronic diseases, it is rarely used in the field of infectious diseases.

We are not aware which article the reviewer mentioned in the comment. If the reviewer could kindly provide the details, we will better respond to your question. We assume that the article you mentioned is the one below.

Xu, M., et al., Association of air pollution with the risk of initial outpatient visits for tuberculosis in Wuhan, China. Occupational and environmental medicine, 2019. 76(8): p. 560-566.

We have at least three differences from this article, as shown below.

First, the models used are different. They used DLNM and we used GAM.

Second, they directly used data from air detection stations, and we used kriging interpolation to predict air pollution before put it an epidemiological model.

Third, we conducted seasonal analysis to see if there is any difference in effects between the cold and warm seasons, whereas this study did not.

Reviewer 3 Report

The study by Huang and colleagues examined the temporal association between the incidence of tuberculosis and several air pollution parameters.

In general, the technical aspects of the study (i.e., the statistical methodology) appear sound. In contrast, consideration of the clinical aspects of the study is conducted in a simplistic manner. I believe that this mismatch has introduced bias and error.

The authors failed to discuss several aspects of the results that appear striking.

(1) It is unclear how “lag” was determined; I was unable to find where it was defined in the manuscript. There are two components to the delay between infection and diagnosis; the delay until the patient seeks medical help and the subsequent delay until confirmation of TB. A systematic review by Sreeramareddy et al. (BMC Infectious Dis 2009;9:91) included several studies from China in which such delays were calculated; the reported total range for both periods was 25 to 71 days. In other words, establishing when a patient was infected is difficult and then correlating this to a lag time for the entire study population in any meaningful way appears impossible. I cannot see the relevance or even the appropriateness of attempting to determine lag time in this type of study (unless the infection date is established). Given that the patient population came from both urban and rural regions (and also covered a wide range of socioeconomic status (SES)), I can imagine that the range could even be greater than reported in the systematic review. The presence and subsequent activation of latent disease could also be a factor – this was not considered.

(2) There was no attempt to adjust the results for potential confounders and, in fact, there appears to be no data on patient clinical or demographic characteristics. Factors such as the already mentioned SES, tobacco use, alcohol or IV drug abuse, and weakened immune systems could all influence delays/lags and susceptibility. Such factors should be provided, examined, and discussed.

(3) The apparent magnitude of effect seems much larger than might be expected; a 10-17% increase in risk. Most air pollution risk increases for the same increase in PM 2.5 used by the authors are much smaller; of the order of “1.0-something” – for example, 1.028 for myocardial infarction (Argacha et al. Int J Cardiol 2016;223:300). Also, the 95% confidence intervals tend to be smaller than those reported in the current study. The authors must discuss why they believe there is such a pronounced association with TB and should also discuss the significance of the large 95% confidence intervals they report. That is, their results must be put in context of other environmental studies.

(4) Similar to point (1) above, the authors should state how the diagnosis of TB was made. In addition, they should then discuss the potential for misclassification.

(5) Instead of using the lag approach to the analysis, it would be more appropriate to examine the potential monthly correlation between PM2.5 and TB. Looking at the gross data, it appears that there is no association; i.e., the months with high levels of PM 2.5 do not correspond to the months when TB is highest. Of course, there can be day-to-day variation; however, if the gross association appears non-existent, why should a reader have faith in a more complex analysis. Such a gross (i.e., monthly) correlation would appear more appropriate because of the uncertainty in date of infection versus date of diagnosis. By the way, the choice of scale for the TB graph (Figure 2 – upper panel) is a poor one. It would be much easier to see the monthly variation if only the box-and-whisker data were presented (and the “dots” omitted – also the figure legend does not explain the format of what is presented).

Author Response

The study by Huang and colleagues examined the temporal association between the incidence of tuberculosis and several air pollution parameters.

In general, the technical aspects of the study (i.e., the statistical methodology) appear sound. In contrast, consideration of the clinical aspects of the study is conducted in a simplistic manner. I believe that this mismatch has introduced bias and error.

Response: Thank you for your insightful comment. Here is our response to your comments.

The authors failed to discuss several aspects of the results that appear striking.

(1) It is unclear how “lag” was determined; I was unable to find where it was defined in the manuscript. There are two components to the delay between infection and diagnosis; the delay until the patient seeks medical help and the subsequent delay until confirmation of TB. A systematic review by Sreeramareddy et al. (BMC Infectious Dis 2009;9:91) included several studies from China in which such delays were calculated; the reported total range for both periods was 25 to 71 days. In other words, establishing when a patient was infected is difficult and then correlating this to a lag time for the entire study population in any meaningful way appears impossible. I cannot see the relevance or even the appropriateness of attempting to determine lag time in this type of study (unless the infection date is established). Given that the patient population came from both urban and rural regions (and also covered a wide range of socioeconomic status (SES)), I can imagine that the range could even be greater than reported in the systematic review. The presence and subsequent activation of latent disease could also be a factor – this was not considered.

Response: Thank you, you have carefully observed the impact of the definition of lag on the results. In this article, we define the lag as the time between when TB cases are recorded to the system and days of exposure prior to their reporting. We have written this definition in the method. As a result, the delay in seeking a doctor and the time it takes to make a diagnosis do affect the results. Nevertheless, this is an inherently unavoidable measurement error that also encountered by many similar studies that examine the effects of short-term exposure to air pollution on infectious diseases. We have listed it as one of our limitations. Here are the changes we made to the article.

"The "lag" is defined as the time between the reporting of a TB case and days of exposure prior to this reporting."

"Fifth, because TB patients may not be able to seek health care immediately after they become ill, and it takes time from the diagnosis to the reporting of a case, the results of this article are affected by this unavoidable measurement error."

(2) There was no attempt to adjust the results for potential confounders and, in fact, there appears to be no data on patient clinical or demographic characteristics. Factors such as the already mentioned SES, tobacco use, alcohol or IV drug abuse, and weakened immune systems could all influence delays/lags and susceptibility. Such factors should be provided, examined, and discussed.

Response: thank you for pointing out that we did not include many confounding factors in our analysis. This is because we do not have such data. We have listed it as another limitation. However, it should be pointed out that one of the advantages of using GAM is that GAM treats a case as its own control, so using GAM can greatly reduce the impact of these confounding factors on the results.

"Fourth, due to the lack of data, some factors that can lead to confounding or effect modification, such as gender, age, socioeconomic status, and so on, have not been included in the analysis, which may result in bias in results."

(3) The apparent magnitude of effect seems much larger than might be expected; a 10-17% increase in risk. Most air pollution risk increases for the same increase in PM 2.5 used by the authors are much smaller; of the order of “1.0-something” – for example, 1.028 for myocardial infarction (Argacha et al. Int J Cardiol 2016;223:300). Also, the 95% confidence intervals tend to be smaller than those reported in the current study. The authors must discuss why they believe there is such a pronounced association with TB and should also discuss the significance of the large 95% confidence intervals they report. That is, their results must be put in context of other environmental studies.

Response: Thank you for your comment. We recalculated the results and found no mistakes. The magnitude of the effect estimate is related to many factors, such as the type of disease, regional difference, the concentration of pollutants, source of pollutants, climate conditions, and so on. For example, the research you mentioned was conducted in Europe with low pollution levels, which may be one of the reasons for the difference in effect estimates between the two studies. Our effect estimates are comparable with another study conducted in Wuhan [1]. For example, their results suggest a 10 μg/m3 increase in NO2 (nitrogen dioxide) was associated with 11.74% (95% CI: 0.70 to 23.98, lag 0–1 weeks), 21.45% (95% CI: 1.44 to 45.41, lag 0–2 weeks) and 12.8% (95% CI: 0.97 to 26.02, lag 0–1 weeks) increase in initial TB consults among all patients with TB.

Xu, M., et al., Association of air pollution with the risk of initial outpatient visits for tuberculosis in Wuhan, China. Occupational and environmental medicine, 2019. 76(8): p. 560-566.

We have added a discussion of the large confidence intervals, as shown below.

"Second, two years of data assumes a small sample size and may reduce the statistical power for the detection of significance. Compared to studies with many years of data, this led to overly wide confidence intervals."

"Considering the above limitations, caution is needed in interpreting the results of the article."

(4) Similar to point (1) above, the authors should state how the diagnosis of TB was made. In addition, they should then discuss the potential for misclassification.

Response: We have added the diagnosis of TB in the methods section, quoted as below.

"In short, a chest X-ray is a common method for detecting tuberculosis, but the following tests are needed to confirm the diagnosis. First, check the patient's sputum for Mycobacterium tuberculosis to confirm the diagnosis. Second, use a lung X-ray examination to diagnose the location, scope, and nature of the disease. Third, conduct a tuberculin test. A positive test indicates the infection."

We have added the discussion for the potential misclassification, quoted below.

"In addition, delays in seeking health care after the onset and diagnosis may also result in delayed reporting from the winter to the following spring [26]. This also led to misclassification of the actual exposure date."

Fares, A., Seasonality of tuberculosis. Journal of global infectious diseases, 2011. 3(1): p. 46.

(5) Instead of using the lag approach to the analysis, it would be more appropriate to examine the potential monthly correlation between PM2.5 and TB. Looking at the gross data, it appears that there is no association; i.e., the months with high levels of PM 2.5 do not correspond to the months when TB is highest. Of course, there can be day-to-day variation; however, if the gross association appears non-existent, why should a reader have faith in a more complex analysis. Such a gross (i.e., monthly) correlation would appear more appropriate because of the uncertainty in date of infection versus date of diagnosis. By the way, the choice of scale for the TB graph (Figure 2 – upper panel) is a poor one. It would be much easier to see the monthly variation if only the box-and-whisker data were presented (and the “dots” omitted – also the figure legend does not explain the format of what is presented).

Response: The relationship between short-term exposure to air pollution and risk of a disease cannot be estimated by observing the raw data, otherwise we can calculate the effect by using linear or non-linear models. The scatter plots presented by the raw data are affected by many factors, such as weather factors, day of the week, long-term effects, and seasonal variations. This is why we use GAM to eliminate these interference factors. In fact many published articles have adopted this method to assess the effect of air pollution on infectious diseases, below are some examples.

Lin, H., Ma, W., Qiu, H., Vaughn, M.G., Nelson, E.J., Qian, Z. and Tian, L., 2016. Is standard deviation of daily PM2. 5 concentration associated with respiratory mortality?. Environmental pollution, 216, pp.208-214. Chen, G., Zhang, W., Li, S., Zhang, Y., Williams, G., Huxley, R., Ren, H., Cao, W. and Guo, Y., 2017. The impact of ambient fine particles on influenza transmission and the modification effects of temperature in China: a multi-city study. Environment international, 98, pp.82-88. Chen, G., Zhang, W., Li, S., Williams, G., Liu, C., Morgan, G.G., Jaakkola, J.J. and Guo, Y., 2017. Is short-term exposure to ambient fine particles associated with measles incidence in China? A multi-city study. Environmental research, 156, pp.306-311. Zhu, S., Xia, L., Wu, J., Chen, S., Chen, F., Zeng, F., Chen, X., Chen, C., Xia, Y., Zhao, X. and Zhang, J., 2018. Ambient air pollutants are associated with newly diagnosed tuberculosis: A time-series study in Chengdu, China. Science of the Total Environment, 631, pp.47-55.

We agree with you that Figure 2 is not a sound figure, which has been removed from the article in light of its minor impact on the main results.

Round 2

Reviewer 1 Report

The manuscript has been much improved. It is good enough to be published. 

Author Response

Thank you for your contributions on our manuscript.

Reviewer 2 Report

Authors have satisfactorily addressed all the concerns in the current version of the manuscript.

Author Response

Thank you for your work. We appreciate your efforts on our manuscript.

Reviewer 3 Report

The study by Huang and colleagues has been improved by the revision. Nevertheless, there are a few remaining issues.

The Introduction never mentions how air pollution exposure could be associated with TB. For readers not familiar with this topic, the initial reaction could well be that I do not understand how air pollution affects bacterial transmission/infection. There needs to be a statement in the Introduction along the lines of, (and this is just an example that comes off the top of my head) “TB may be associated with increased exposure to air pollution because; (1) components of the pollution may weaken the immune system by affecting T-cells, macrophages… (2) there may be damage to epithelial cells in the respiratory tract or to the lung caused by oxidative stress induced by air pollution, or (3) direct transport of bacteria on pollutant particles.

Obviously, what I have written may not be phrased appropriately. However, if something like this is included in the Introduction, it will make the entire paper more readable (and less of a mystery) for non-experts.

A couple of recent references that might be worthwhile including to support these statements are; Torres et al. Thorax 2019;74:675 & Popovic et al. Environ Res 2019;170:33-45.

Am I correct in thinking that Kriging interpolation assumes isotropy? This would mean that it would not take into account the built environment for example. This is something to add to the limitations section. I appreciate the approach is likely better than not using it; however, Kriging is not perfect and, for those unfamiliar with the technique, some objective assessment of the limitations would be appropriate.

There is an apparent disconnect between the map (Figure 1) and the text. The text states 51 monitoring sites were included “with 10 in Wuhan”. I could only see 7 (?) stars on the maps. What happened to the other sites? Either more stars should be added or a more detailed explanation should be included in the figure legend.

Because, as the authors acknowledge in their response, the delay between infection and seeking treatment is highly variable, were the authors surprised at the consistency of the seven day lag? To me, this consistency seems very surprising. This issue should be discussed.

Author Response

The study by Huang and colleagues has been improved by the revision. Nevertheless, there are a few remaining issues.

Responses: Thank you for your comments on this article. The following are our responses to your remaining concerns.

The Introduction never mentions how air pollution exposure could be associated with TB. For readers not familiar with this topic, the initial reaction could well be that I do not understand how air pollution affects bacterial transmission/infection. There needs to be a statement in the Introduction along the lines of, (and this is just an example that comes off the top of my head) “TB may be associated with increased exposure to air pollution because; (1) components of the pollution may weaken the immune system by affecting T-cells, macrophages… (2) there may be damage to epithelial cells in the respiratory tract or to the lung caused by oxidative stress induced by air pollution, or (3) direct transport of bacteria on pollutant particles.

Obviously, what I have written may not be phrased appropriately. However, if something like this is included in the Introduction, it will make the entire paper more readable (and less of a mystery) for non-experts.

A couple of recent references that might be worthwhile including to support these statements are; Torres et al. Thorax 2019;74:675 & Popovic et al. Environ Res 2019;170:33-45.

Responses: We have added several sentences at the end of paragraph 2 in the introduction section to explain the mechanisms.

"There are some possible mechanisms to explain the link between air pollution and the increased risk of TB. These mechanisms include: (1) both gaseous pollutants and PM may trigger the disease by weakening the immune system (e.g., affecting T cells or impairing macrophage function); (2) oxidative stress and inflammatory reactions induced by air pollution may result in damage to respiratory tract or lung epithelial cells; and (3) the direct transport of bacteria and attached to particles make healthy population infected. These mechanisms are discussed in more detail in the discussion section."

References:

  1. Torres, M., Carranza, C., Sarkar, S., Gonzalez, Y., Vargas, A.O., Black, K., Meng, Q., Quintana-Belmares, R., Hernandez, M., Garcia, J.J.F.A. and Páramo-Figueroa, V.H., 2019. Urban airborne particle exposure impairs human lung and blood Mycobacterium tuberculosis immunity. Thorax, 74(7), pp.675-683.
  2. Popovic, I., Magalhaes, R.J.S., Ge, E., Marks, G.B., Dong, G.H., Wei, X. and Knibbs, L.D., 2019. A systematic literature review and critical appraisal of epidemiological studies on outdoor air pollution and tuberculosis outcomes. Environmental research, 170, pp.33-45.

Am I correct in thinking that Kriging interpolation assumes isotropy? This would mean that it would not take into account the built environment for example. This is something to add to the limitations section. I appreciate the approach is likely better than not using it; however, Kriging is not perfect and, for those unfamiliar with the technique, some objective assessment of the limitations would be appropriate.

Responses: The reviewer is right. The kriging does assume isotropy and our method did not take the built environment into consideration. We have listed it as a limitation of kriging in the discussion section.

"Furthermore, the kriging also assumes isotropy, that is, it assumes uniformity in all directions. This obviously ignores the impact of the real environment (such as the built environment) on exposure."

Reference:

  1. Remy, N., Boucher, A. and Wu, J., 2009. Applied geostatistics with SGeMS: A user's guide. Cambridge University Press.

There is an apparent disconnect between the map (Figure 1) and the text. The text states 51 monitoring sites were included “with 10 in Wuhan”. I could only see 7 (?) stars on the maps. What happened to the other sites? Either more stars should be added or a more detailed explanation should be included in the figure legend.

Responses: Figure 1 actually shows 10 stars, we re-counted and found no omissions. It may be because there are several stars on the outer edge of the city so the reviewer can hardly see them.

Because, as the authors acknowledge in their response, the delay between infection and seeking treatment is highly variable, were the authors surprised at the consistency of the seven day lag? To me, this consistency seems very surprising. This issue should be discussed.

Responses: Thank you for your question. Our results show that the effect of each pollutant on TB reaches its maximum at lag 7, which may be due to collinearity between the pollutants. In other words, the effect of any pollutant on TB actually reflects the effect of another or a combination of several pollutants on TB. We think this consistency does not have much to do with the delay between infection and the hospital visit, as the delay will only result in a shift of concentration for all pollutants in units of days. The concentration of any pollutant will not change compared to other pollutants on a given day. Therefore, we only added the explanation of this consistency due to collinearity in the discussion section, quoted below.

"The surprising consistency observed in lag 7 may be due to the ubiquitous collinearity among the pollutants, that is, the effect of any pollutant on TB actually reflects the effect of another or a combination of several pollutants on TB."